# How does Watermarking Affect Visual Language Models in Document Understanding?

**Chunxue Xu**[†], **Yiwei Wang**[§], **Bryan Hooi**[‡], **Yujun Cai**[◇], **Songze Li**[†*]
[†] Southeast University, China    [§] University of California, Merced, USA
[‡] National University of Singapore, Singapore
[◇] The University of Queensland, Australia

## Abstract

Visual Language Models (VLMs) have become foundational models for document understanding tasks, widely used in the processing of complex multimodal documents across domains such as finance, law, and academia. However, documents often contain noise-like information, such as watermarks, which inevitably leads us to inquire: *Do watermarks degrade the performance of VLMs in document understanding?* To address this, we propose a novel evaluation framework to investigate the effect of visible watermarks on VLMs performance. We takes into account various factors, including different types of document data, the positions of watermarks within documents and variations in watermark content. Our experimental results reveal that VLMs performance can be significantly compromised by watermarks, with performance drop rates reaching up to 36%. We discover that *scattered* watermarks cause stronger interference than centralized ones, and that *semantic contents* in watermarks creates greater disruption than simple visual occlusion. Through attention mechanism analysis and embedding similarity examination, we find that the performance drops are mainly attributed to that watermarks 1) force widespread attention redistribution, and 2) alter semantic representation in the embedding space. Our research not only highlights significant challenges in deploying VLMs for document understanding, but also provides insights towards developing robust inference mechanisms on watermarked documents.

## 1 Introduction

The rapid advancement of Visual Language Models (VLMs) has transformed document understanding capabilities, enabling sophisticated processing of complex multimodal content without relying on traditional Optical Character Recognition (OCR) technologies. Unlike conventional document processing systems that convert text content into machine-readable characters (Srihari, 1986; Hwang et al., 2021; Hong et al., 2022), modern VLMs directly process original document images, seamlessly integrating visual features with textual information. This OCR-free approach eliminates potential recognition errors in documents with complex layouts, non-standard fonts, and visual noise (Kim et al., 2022; Liu et al., 2024b; Hu et al., 2024), while preserving critical layout and semantic structures that enhance performance in tasks such as question answering, information retrieval, and cross-page analysis.

However, this advancement introduces new robustness challenges that remain largely unexplored. While considerable research has examined the vulnerability of VLMs to adversarial inputs and visual noise in general contexts, the specific impact of common document modifications, particularly watermarks on VLMs performance, represents a critical gap in our understanding. Unlike Large Language Models (LLMs), where robustness mechanisms against harmful queries are relatively well-established (Mei et al., 2024), VLMs' multimodal

---

*Corresponding Author.

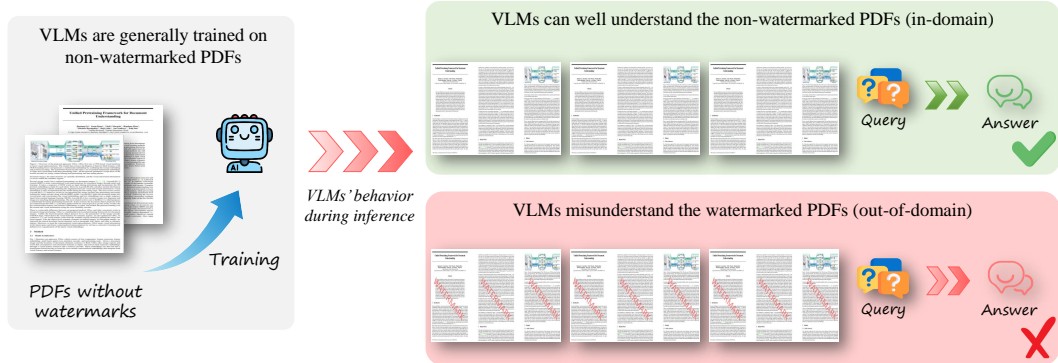

Figure 1: After being fine-tuned on unwatermarked document data, the LVLM provides correct responses to question with unwatermaked document but exhibits obvious errors when responding to watermarked one.

nature creates unique vulnerability surfaces where image features can potentially bypass or undermine existing safeguards.

Watermarking represents one of the most prevalent forms of visual modification in professional documents, serving as a standard method for copyright protection and document authentication (Kankanhalli et al., 1999). As illustrated in Figure 1, even a simple watermark can dramatically alter an VLMs' document interpretation, causing the model to produce incorrect responses despite no changes to the underlying informational content. This observation raises fundamental questions about VLMs reliability in real-world document processing scenarios, where watermarked materials are the norm rather than the exception.

To address this critical knowledge gap, we introduce a comprehensive evaluation framework specifically designed to assess VLMs robustness against watermarks in document understanding tasks. Our approach considers the unique characteristics of document-based question answering across diverse content types, including text-heavy documents, charts, and tables (Suri et al., 2024). By systematically embedding visible watermarks with varying properties into clean documents, we generate a watermarked dataset and the original clean dataset. Each dataset entry comprises a document image [1] accompanied by a corresponding question. Our evaluation is conducted on four state-of-the-art VLMs. The model performance is assessed across the two datasets, providing insights into the model's sensitivity to visible watermarks. The primary objectives of our study are as follows: ❶ To analyze the impact of visible watermarks on the performance of VLMs for document understanding. ❷ To identify the key characteristics of visible watermarks that most significantly influence model robustness.

With the proposed framework, we perform numerous controlled experiments to find that increasing watermark size does not necessarily amplify the interference effect, contradicting our initial hypothesis. Further analysis reveals that: (1) For watermarks with the same total area, distributing them across multiple locations results in significantly stronger interference than placing them in a single fixed position; (2) By examining watermarked documents that successfully cause perturbation, we observe that effective watermarks are often located near the region containing the standard answer, suggesting that obstruction of key content leads to interference; (3) To validate this 'occlusion' reason, we replace text watermark 'MARK' with rectangular mask to improve the probability of covering key information and repeat the experiments. However, the interference effect is notably weaker compared to text-based watermarks, highlighting the importance of *semantic information* in watermarks rather than 'occlusion' and provide insights for designing more robust watermarking strategies.

We analyze the impact of watermark position and content from two perspectives: attention mechanisms and semantic similarity in embedding space. Our findings show that: (1) The

---

[1]Documents exist in various formats, such as DOC, PDF, and PPT, however, images provide a standardized representation.

most disruptive watermark position causes the widest redistribution of attention weights. (2) The most disruptive watermark content results in the lowest semantic similarity between the embeddings of watermarked and original documents.

This paper makes several contributions to the literature: First, we systematically study and evaluate the vulnerabilities of VLMs for document understanding for the first time. Second, we propose a new framework to evaluate the impact of visual watermarks on VLMs performance. Third, we analyze how watermark position and content affect VLMs performance, and use visualization to reveal the underlying causes.

## 2 Related Works

### 2.1 Visual Language Models for Document Understanding

The field of visual document understanding focuses on understanding structure in visual documents and extracting key information to convert structured or semi-structured forms into machine-readable formats. A visual document is a document that contains a variety of visual and text elements. These documents integrate a variety of visual elements, including paragraphs, charts, tables, etc. Text elements (such as paragraphs and lists) and visual elements (such as charts and tables) together constitute the semantics of the documents. QA is one of the key tasks in visual document understanding, which involves answering natural language questions based on the context information of visual documents(Ding et al., 2024).

With the rapid development of visual document understanding requirements, VLMs technology for visual document understanding is introduced: Qwen-VL (Bai et al., 2023) retains the Q-Former architecture while replacing the original language model with a larger one. In order to extend the capability of VLMs to visual document understanding, mPLUG-DocOwl (Ye et al., 2023a) proposes a modular model based on mPLUG-Owl (Ye et al., 2023b) for document understanding without OCR. These comprehensive approaches highlight the potential of VLMs in handling complex visual language tasks (Li et al., 2024).

### 2.2 Robustness of Visual Language Models

With the widespread application and impressive performance of VLMs in multimodal QA and reasoning tasks, their robustness has garnered increasing attention in recent years. Zhao et al. (2023) evaluate the adversarial robustness of VLMs and find that multimodal generation increases security risks and that an attacker can deceive the model by manipulating visual input. Qiu et al. (2024) observe that the multimodal model is not robust to image and text perturbation, especially to image perturbation. In the perturbation method tested, character-level perturbation causes the most serious distribution deviation of text, and zoom blur is the most serious deviation of image data. Lee et al. (2024) introduce VHELM to evaluate the robustness of VLMs through a variety of test scenarios, such as introducing typos and testing robustness to sketches and out-of-distribution images. Agarwal et al. (2025) introduce MVTamperBench, designed to evaluate the robustness of VLMs against video tampering effects such as rotation, occlusion, substitution, and repetition.

Significant progress has been achieved in robustness research of VLMs. In this work, we focus on how visual watermarks influence VLMs performance when applied to document understanding, identifying potential risks in visual question answering scenarios.

## 3 Methodology

In this section, we describe our methodology for evaluating the impact of watermarks on VLMs in document understanding tasks. We first formalize the document understanding task and define performance metrics to quantify model accuracy. Following this, we detail our evaluation datasets and experimental setup, including the selection of models and watermark parameters.

## 3.1 Document Understanding Task and VLMs Performance Metric

We use visual question answering (VQA) as our primary task. Because VQA is the most essential and widely used task in document understanding scenarios. (Li et al., 2023). The VQA task is defined as follows:

Given a user question $q$ and a document image $i$ as visual context, the system is tasked with generating an answer. Each question is accompanied by a corresponding ground truth $gt$, which serves as the reference for evaluation. Our evaluation dataset $\mathcal{D}_{\text{eval}}$ consists of representative VQA data, with each sample structured as $(q, i, gt)$. For a given VLM model $f$, which takes the question-image pair $(q, i)$ as input and generates an answer, the *answer accuracy* over $\mathcal{D}_{\text{eval}}$ is defined as:

$$\text{Acc}(f, \mathcal{D}_{\text{eval}}) \overset{def}{=} \frac{1}{|\mathcal{D}_{\text{eval}}|} \sum_{(q,i,gt) \in \mathcal{D}_{\text{eval}}} \mathbb{1}(f(q,i), gt),$$

where $\mathbb{1}(\cdot)$ is the indicator function, which takes the value of 1 if the model's generated answer $f(q, i)$ matches $gt$, and 0 otherwise.

To evaluate the impact of the watermark on the model's performance, we inject a watermark $w$ into each input image $i$ in $\mathcal{D}_{\text{eval}}$, resulting in a perturbed evaluation dataset, denoted as $\mathcal{D}'_{\text{eval}}$, which consists of samples in the form of $(q, i + w, gt)$.

To quantify this impact, we adopt the **Performance Drop Rate (PDR)** metric (Zhu et al., 2024), which measures the percentage decrease in *answer accuracy* with respect to the user question $q$. The PDR of VLM $f$ under the influence of the watermark is defined as:

$$\text{PDR}(f, \mathcal{D}_{\text{eval}}, \mathcal{D}'_{\text{eval}}) \overset{def}{=} \frac{\text{Acc}(f, \mathcal{D}_{\text{eval}}) - \text{Acc}(f, \mathcal{D}'_{\text{eval}})}{\text{Acc}(f, \mathcal{D}_{\text{eval}})}.$$

The PDR metric directly captures the degradation in model performance when processing watermarked documents compared to original documents, providing a clear measure of how significantly watermarks interfere with the model's document understanding capabilities. A higher PDR value indicates a stronger disruptive effect of watermarks on the model.

## 3.2 Experimental Setting

**Evaluation Dataset**  We construct our datasets by randomly sampling 100 images for each of the three document types — text, chart, and table — from the public mPLUG-DocOwl2 dataset (Hu et al., 2024) and the LongDocURL dataset available on Hugging Face (Dengchao, 2024). These images are categorized into four classes: TextS, ChartS, ChartM, and TableS, corresponding to text document images, chart document images, and table document images, along with their respective question types (single choice or multiple response).

For each image, we design a set of questions based on its content, providing four answer choices for the model to select from. The number of correct answers is determined by the category of the image: images in the ChartM class have questions with multiple correct answers, whereas images in the TextS, ChartS, and TableS classes have questions requiring only a single correct choice. The resulting four datasets are as follows:

- **TextS Set**: Each instance in this dataset consists of a text document image paired with a corresponding single-choice question. Each question presents four answer options, and the groundtruth is a single correct option among them.
- **ChartS Set**: Each instance in this dataset consists of a chart document image paired with a corresponding single-choice question. Each question includes four answer choices, with only one correct answer designated as the groundtruth.
- **ChartM Set**: Each instance in this set consists of a chart document image paired with a multiple-response question. Each question provides four answer choices, and the groundtruth comprises more than one correct options among them.

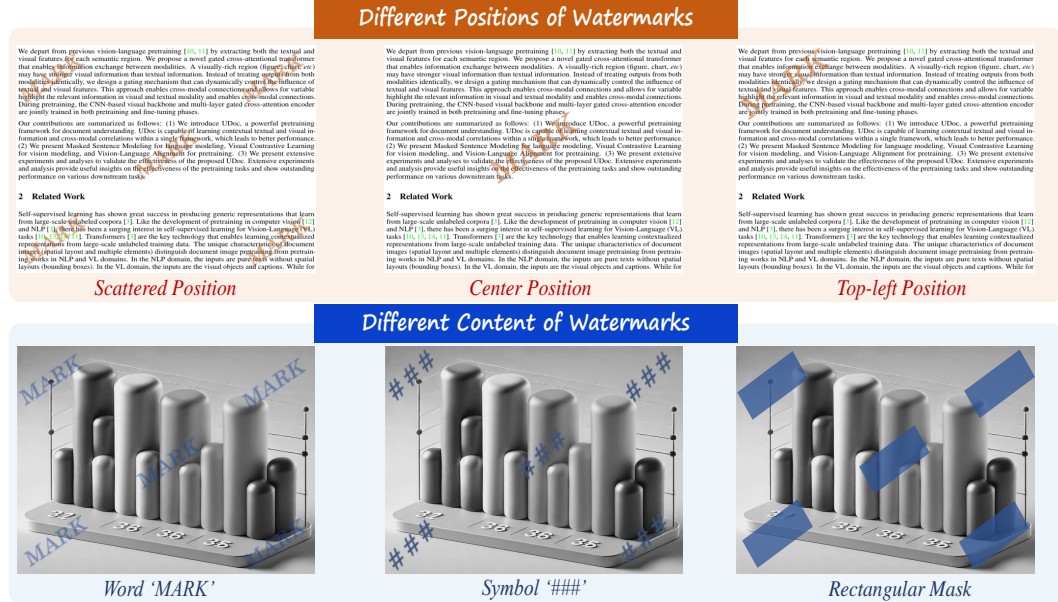

Figure 2: **Position and content exploration.** (a) For position exploration, the watermark is placed Scattered in five fixed positions, in Center and in Top-left corner of document image; (b) For content exploration, we set the watermark content to text 'MARK', '###', and fill rectangular box where the watermark is located with a rectangular Mask.

- **TableS Set**: Each instance in this set consists of a table document image paired with a single-choice question. Each question contains four possible answers, with only one correct option specified as the groundtruth.

For text-based documents, 50% are from real estate and finance, and 50% from law-related studies. Chart and table documents are extracted from academic papers in computer science, physics, and related fields. This diverse collection of document types and question formats enables us to comprehensively evaluate how watermarks affect VLMs across different document understanding scenarios, from simple text extraction to complex chart interpretation and tabular data analysis.

**Evaluation Models.** We select 4 representative sate-of-the-art VLMs for our evaluation: mPLUG-DocOwl2 (Hu et al., 2024) is designed specifically for document understanding tasks, employing a specialized architecture for processing document layouts and mixed content types.
InternVL-2.5 (Chen et al., 2025) excels in real-world understanding benchmarks and is capable of handling high-resolution images and complex document layouts.
LLaVA-v1.5 (Liu et al., 2024a) features an attention mechanism that dynamically focuses on key parts of text or images while processing document content.
Qwen2.5-VL (Bai et al., 2025) demonstrates strong performance in document understanding tasks, particularly in structured data extraction and long document understanding.
This selection covers a range of model architectures and specializations, allowing us to identify both general trends and model-specific behaviors in response to watermarked documents.

**Experimental Parameters.** We primarily investigate the impact of watermark position and content on the interference effect. All watermarks are programmatically added to the document images using the Python Pillow library. To ensure experimental consistency and isolate the effects of other watermark properties, we keep several key parameters fixed: (1) We set watermark color as black and set the watermark transparency to 0.5 to ensure that the existence of the watermarks does not affect the visibility of the document to humans,

Table 1: **PDR**(%) **for different watermark positions**. Note: The pink numbers represent the highest PDR value across different watermark positions.

| Model | TextS | | | ChartS | | | ChartM | | | TableS | | |
|---|---|---|---|---|---|---|---|---|---|---|---|---|
| | Center | Scattered | Top-left | Center | Scattered | Top-left | Center | Scattered | Top-left | Center | Scattered | Top-left |
| mPLUG-DocOwl2 | 12 | 14 | 1 | 1 | 11 | 3 | 15 | 24 | 9 | 24 | 28 | 1 |
| InternVL-2.5-8b | 21 | 28 | 16 | 2 | 2 | 1 | 25 | 36 | 28 | 10 | 11 | 14 |
| LLaVA-v1.5-7b | 19 | 9 | 11 | 9 | 23 | 20 | 16 | 22 | 21 | 21 | 17 | 10 |
| Qwen2.5-VL-7b | 0 | 0 | 0 | 4 | 6 | 5 | 1 | 7 | 2 | 0 | 0 | 0 |
| AVG | 13 | 12 | 7 | 4 | 10 | 7 | 14 | 22 | 15 | 13 | 14 | 6 |

but also has a certain impact on the performance of the model, which is more in line with the realistic scene of document watermarks; (2) We fix the watermark angle at 0 degrees to eliminate orientation as a variable; (3) Additionally, in each experimental condition, the proportion of watermarks in the total area of the document image is distributed between 10% and 80%. Experiments on watermark angle and opacity are included in the appendix.

**Considered Watermark Properties.** Our experiments focus on two primary dimensions of watermark properties, examining how each affects VLMs performance (see Figure 2):
(1) **Watermark Position**: We systematically investigate how watermark placement influences VLMs performance by positioning watermarks in three distinct configurations. The Center placement positions the watermark at the center of the document image. The Top-left configuration places the watermark in the top-left corner of the document. The Scattered approach distributes watermarks across five fixed positions throughout the document.
(2) **Watermark Content**: We evaluate how different types of watermark content affect VLMs performance using three variants. The MARK content consists of a text-based watermark containing the word 'MARK'. The '###' uses a symbol-based watermark with the pattern '###'. The Mask variant employs a rectangular mask placed in the same location as the text-based watermark.
For both dimensions, we carefully control all variables except the one being explored. This ensures that any observed differences in performance can be directly attributed to the specific watermark property under investigation. The total watermark area, color, transparency, and other factors remain consistent across experimental conditions within each dimension.

# 4 Impact of Document Watermarks on VLMs Performance

Table 1 and Table 2 show VLMs $\text{PDR}(f, \mathcal{D}_{\text{eval}}, \mathcal{D}'_{\text{eval}})$ on datasets with watermarks of different positions and contents. In general, the highest PDR can reach 36%, indicating that VLMs are susceptible to the interference of document watermarks when processing document understanding tasks.

What's more, we investigate the impact of watermark color and area ratio on model performance. Results (see appendix) show that: (1) Color change doesn't significantly affect VLMs performance. This observation can be attributed to how modern VLMs process textual and visual information. These models primarily focus on high-level semantic understanding rather than low-level pixel variations. (2) A larger watermark area ratio does not necessarily increase interference. This is because when the watermark area is too large, the semantic information carried by the watermark becomes diluted, making it ineffective in interfering with document information.

## 4.1 Impact of Watermark Position

### 4.1.1 PDR across Watermark Positions

We can observe from Table 1 that the interference effect of watermarks is significantly influenced by their position.

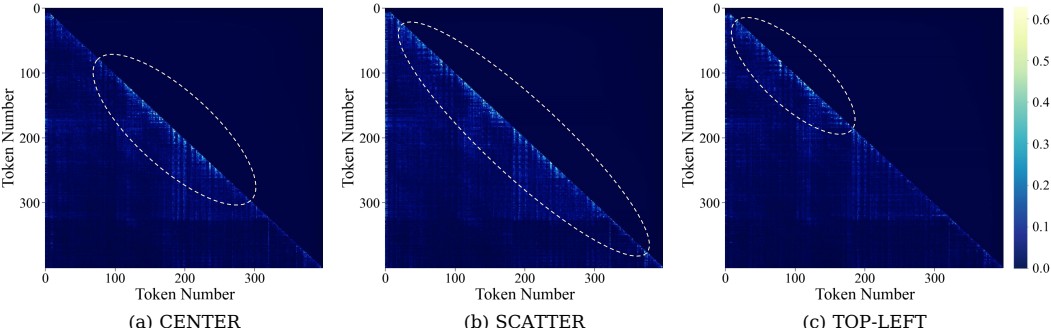

Figure 3: **Heatmap of attention wight variation** between watermarked multimodal input sequences and original one. `Brighter points` indicate the model's attention weights for the corresponding tokens have changed, with higher brightness indicating greater changes. The `yellow dashed line` indicates the region where the attention weight changes most significantly.

(1) **Position Effect.** Watermarks of SCATTERED configuration consistently result in higher PDR values across most testing cases compared to those of the CENTER and TOP-LEFT configurations. In 11/16 cases, the SCATTERED position yields the highest PDR among the three locations, demonstrating its dominant impact. For MPLUG-DOCOWL2 on the CHARTM dataset, SCATTERED watermarks achieve 24% PDR compared to 15% for CENTER and 9% for TOP-LEFT placements. This pattern is particularly evident in the CHARTS and CHARTM datasets, where SCATTERED watermarks consistently lead to the highest PDR values across all tested models.

(2) **Dataset Vulnerability.** The overall PDR levels vary significantly across datasets. The CHARTM dataset exhibits the highest PDR values, where it reaches up to 36% (INTERNVL-2.5 of SCATTERED configuration). This suggests that the CHARTM dataset, which by nature allows for multiple correct answers, already presents a challenge to model performance. The presence of watermarks exacerbates this issue, further reducing the model's prediction accuracy.

(3) **Model Robustness.** Different VLMs exhibit varying sensitivity to watermarks. QWEN2.5-VL consistently demonstrates low PDR values, with PDR values even dropping to zero on the TEXTS and TABLES datasets, indicating its strong robustness against watermarks. In contrast, INTERNVL-2.5 and MPLUG-DOCOWL2 show higher vulnerability to watermark interference, especially on chart-based documents.

### 4.1.2 Attention Mechanism Analysis

To further explore the underlying reasons why watermarks with different positions produce different interference effects, we calculate the variation in *attention weights* generated by MPLUG-DOCOWL2 model between the watermarked multimodal input sequence and the non-watermarked input sequence, and use it to produce the attention weight variation heatmap (see Figure 3). The same analysis is also conducted for QWEN2.5-VL and LLAVA-v1.5, with results in the appendix. The visualization reveals distinct patterns: the model adjusts *attention weights* across the entire image under the SCATTERED setting (see Figure 3(b)), while attention changes are concentrated in the middle (see Figure 3(a)) and beginning portions (see Figure 3(c)) of the sequence under the CENTER and TOP-LEFT settings, respectively.

These attention variations can be attributed to the VLM's processing architecture. Before performing multimodal fusion, the model divides the image into different patches. When watermarks are embedded in different positions, they affect different patches in the input sequence, altering the embedding vectors at these specific locations. These changes subsequently induce corresponding modifications in the *attention weights* at these positions.

The key insight from these results is that SCATTERED watermarks disrupt the model's attention across multiple regions, forcing widespread *attention redistribution* that compromises the

Table 2: **PDR(%) for different watermark content**. Note: The purple numbers represent the highest PDR value across different watermark content.

| Model | TextS | | | ChartS | | | ChartM | | | TableS | | |
|---|---|---|---|---|---|---|---|---|---|---|---|---|
| | 'MARK' | '###' | Mask | 'MARK' | '###' | Mask | 'MARK' | '###' | Mask | 'MARK' | '###' | Mask |
| mPLUG-DocOwl2 | 6 | 8 | 16 | 11 | 6 | 1 | 28 | 10 | 9 | 33 | 32 | 7 |
| InternVL-2.5-8b | 22 | 25 | 17 | 2 | 1 | 1 | 35 | 26 | 28 | 14 | 9 | 11 |
| LLaVA-v1.5-7b | 16 | 21 | 3 | 22 | 17 | 13 | 12 | 23 | 23 | 18 | 16 | 14 |
| Qwen2.5-VL-7b | 0 | 0 | 0 | 6 | 8 | 0 | 2 | 5 | 2 | 0 | 0 | 0 |
| AVG | 11 | 13 | 9 | 10 | 8 | 4 | 19 | 16 | 16 | 16 | 14 | 8 |

model's ability to maintain coherent document understanding. In contrast, when watermarks are concentrated in a single location (CENTER or TOP-LEFT), the model primarily adjusts its attention in that affected region while preserving its processing of information elsewhere. This localized adaptation allows the model's *semantic understanding* to remain largely intact, resulting in lower performance degradation compared to the SCATTERED configuration.

## 4.2 Impact of WATERMARK CONTENT

After examining position effects, we now investigate how different watermark content types affect VLMs performance. While position determines where interference occurs, content characteristics may influence how the model processes and interprets the watermarked information. We test three distinct watermark content: text-based ('MARK'), symbol-based ('###'), and visual mask (MASK) to understand their varying impacts on document understanding capabilities.

### 4.2.1 PDR across Watermark Content

As shown in Table 2, watermark content significantly influences interference effects across models and datasets.

(1) **Content Effect.** When the watermark content is either 'MARK' or '###', the PDR values of VLMs are generally higher compared to the MASK watermarks. From a statistical perspective, the average PDR across all models and datasets for the 'MARK' setting reaches 14%, for the '###' setting 13%, and for the MASK setting only 9%. In most cases, the PDR under the 'MARK' setting are highly similar to that under the '###' setting, while the MASK setting exhibits a noticeably lower and more varied interference effect. These results highlight that different watermark content types impact document understanding tasks to varying degrees, with explicit textual and symbolic watermarks being the most disruptive across multiple models and datasets.

(2) **Model Robustness.** Different VLMs show varying sensitivity patterns to watermark content. InternVL-2.5 and LLaVA-v1.5 are more susceptible to interference from '###' watermarks on the TextS dataset, whereas they are more affected by 'MARK' watermarks on other datasets. Similarly, the Qwen2.5-VL model shows sensitivity to watermarks only on the ChartS and ChartM datasets, with the highest PDR in the '###' configuration. Additionally, the mPLUG-DocOwl2 model exhibits significant sensitivity to MASK watermarks on the TextS dataset, with its PDR values for 'MARK' and '###' watermarks being substantially lower than those for MASK.

### 4.2.2 Semantic Interference Analysis

The experimental results indicate that the impact of watermarks on model performance is not solely due to their obstruction of document information. If the primary effect were merely occlusion, the MASK watermarks should outperform the 'MARK' and '###' watermarks. However, our findings do not fully support this assumption. Therefore, we further analyze the influence of watermark content from the perspective of *semantic alterations* in the *cross-modal embedding* of input sequence.

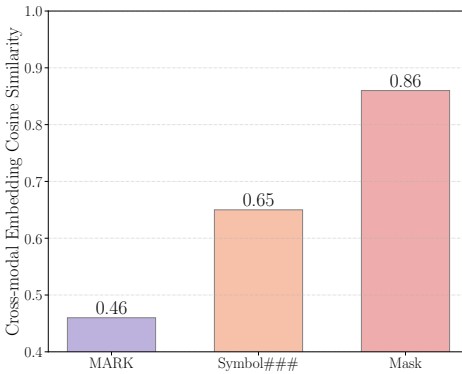

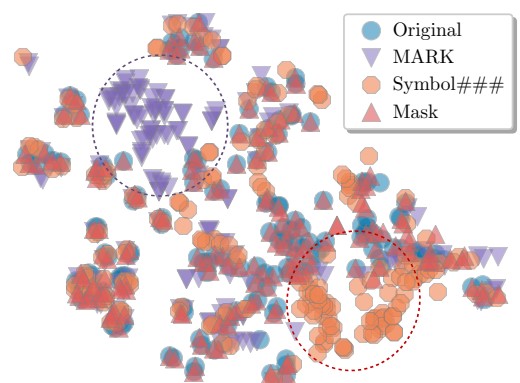

Figure 4: **Cross-modal embedding cosine similarity** of watermarked input sequences with the original one.

Figure 5: Dimentional **t-SNE visualization** of input cross-modal embeddings with watermarks in different content.

**Cross-modal Embedding Cosine Similarity.** *Cosine similarity* evaluates the similarity between two embedding vectors in the semantic space by measuring the directional similarity between them (Zhu et al., 2025; Salton et al., 1975). The higher the cosine similarity score is, the more similar the two embedding vectors are. We compute the cosine similarity between the *cross-modal embedding* vectors obtained from the watermarked and non-watermarked input sequence using the mPLUG-DocOwl2 model (see Figure 4). The results reveal that the 'MARK' watermark leads to the lowest cosine similarity with the original input (0.46), whereas the Mask watermark results in the highest similarity (0.86). This observation further confirms that although the Mask watermark maximizes the occlusion of key information, it induces minimal changes in the cross-modal embeddings' semantic information. Consequently, this can explain why the average PDR value under the Mask setting is the lowest.

**Dimentional t-SNE Visualization for Cross-modal Embedding.** t-SNE (Hinton & Roweis, 2002) can be used to compare the distribution characteristics of *cross-modal embedding*s before and after watermarking. If the addition of watermarks causes a significant change in the embedding distribution, it indicates that watermarks have affected the model's semantic understanding. As shown in Figure 5 , the projections of *cross-modal embedding*s (generated from mPLUG-DocOwl2) with 'MARK' and '###' watermarks in the two-dimensional space exhibit a noticeable deviation from the original input data points, whereas embedding with Mask watermarks remain highly consistent with the original one, showing almost no significant change. This phenomenon suggests that 'MARK' and '###' watermarks introduce additional explicit information, altering semantic structure of the input content, which in turn leads to a substantial shift in the *cross-modal embedding*. In contrast, although Mask watermarks visually obscure part of the input content, they don't add new semantic information. As a result, the model can maintain a relatively stable embedding distribution during cross-modal alignment. This also explains why the PDR values for 'MARK' and '###' watermark settings are more similar, while the PDR for the Mask watermark differs certainly from the other two.

## 5   Conclusion

In this paper, we establish a framework based on VQA datasets to systematically evaluate the robustness of VLMs in document understanding task for the first time. Through comprehensive experiments on four state-of-the-art VLMs, we observe that watermarks in documents can introduce varying degrees of interference in model performance. Our study provides new insights into robustness evaluation for VLMs in document understanding tasks, highlighting the need for further research in assessing and enhancing the robustness of VLMs in multimodal document processing.

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

# A  Appendix

## A.1  Impact of Watermark Color

We compare the PDR across datasets with different watermark colors on mPLUG-DocOwl2, all positioned in a Scattered position. The results, shown in Table 3, indicate that average PDR scores across different colors are similar. This observation can be attributed to how modern VLMs process textual and visual information. These models primarily focus on high-level semantic understanding rather than low-level pixel variations. As a result, changes in watermark color do not significantly affect the model's interpretation of the document, leading to minimal differences in interference effects.

Table 3: **Average PDR(%) for different watermark colors on mPLUG-DocOwl2.**

| Color | TextS | | | ChartS | | | ChartM | | | TableS | | | AVG |
|---|---|---|---|---|---|---|---|---|---|---|---|---|---|
| | 'MARK' | '###' | Mask | 'MARK' | '###' | Mask | 'MARK' | '###' | Mask | 'MARK' | '###' | Mask | |
| Black | 11 | 7 | 21 | 20 | 10 | 2 | 55 | 9 | 5 | 39 | 38 | 76 | 19 |
| Red | 0 | 0 | 32 | 18 | 10 | 1 | 63 | 33 | 4 | 31 | 35 | 4 | 20 |
| Green | 0 | 1 | 30 | 7 | 11 | 0 | 64 | 20 | 2 | 30 | 40 | 3 | 22 |

## A.2  Impact of Watermark Area Ratio

As shown in Figure 6, increasing the area ratio of the watermark does not necessarily lead to a stronger interference effect. This suggests that once the watermark area exceeds a certain threshold, it is no longer the decisive factor influencing model performance. This can be attributed to that when the watermark area is too large, the semantic information carried by the watermark becomes diluted, making it ineffective in interfering with the original content of the document.

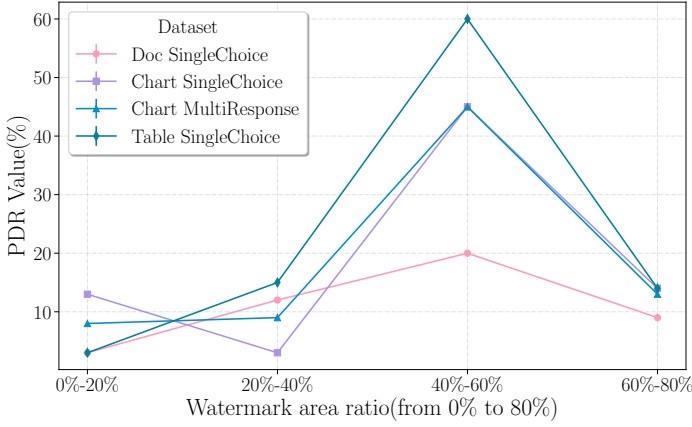

Figure 6: **Average PDR(%) of different watermark area ratios** on the full dataset.

## A.3  Impact of Watermark Opacity

As shown in Table 4, increasing the opacity of the watermark leads to a significant rise in performance degradation, with the average PDR increasing from 8.5% at low opacity ($\alpha = 0.2$) to 21.3% at high opacity ($\alpha = 0.8$). This trend suggests that higher visual prominence caused by denser watermark opacity results in stronger disruption of model attention. The underlying reason may be that more opaque watermarks dominate the visual

space, increasing misalignment in the model's attention mechanism and impairing its ability to focus on relevant document content.

Table 4: Average PDR(%) for different watermark opacity levels on mPLUG-DocOwl2.

| Opacity ($\alpha$) | Avg. PDR (%) |
|---|---|
| 0.2 (light) | 8.5 |
| 0.5 (medium) | 15.7 |
| 0.8 (dense) | 21.3 |

## A.4  Impact of Watermark Rotation

As shown in Table 5, variations in watermark rotation angle have minimal effect on model performance. The average PDR remains relatively stable across different angles, with 14.2% at 0°, 14.5% at 45°, and 14.1% at 90°. This consistency indicates that modern VLMs exhibit strong robustness to moderate geometric transformations in watermark orientation. A possible explanation is that the spatial encoding mechanisms of current vision backbones effectively normalize such rotation variations, making them less disruptive to cross-modal alignment.

Table 5: Average PDR(%) for different watermark rotation angles on mPLUG-DocOwl2.

| Rotation Angle | Avg. PDR (%) |
|---|---|
| 0° | 14.2 |
| 45° (diagonal) | 14.5 |
| 90° (vertical) | 14.1 |

## A.5  Attention Heatmap Analysis Across VLMs

As shown in the Figure 7 and Figure 8, these are the attention variation heatmaps of Qwen2.5-VL and LLaVA-v1.5. It is evident that the Qwen2.5-VL exhibits stronger robustness compared to LLaVA-v1.5 and mPLUG-DocOwl2, which is consistent with the experimental results.

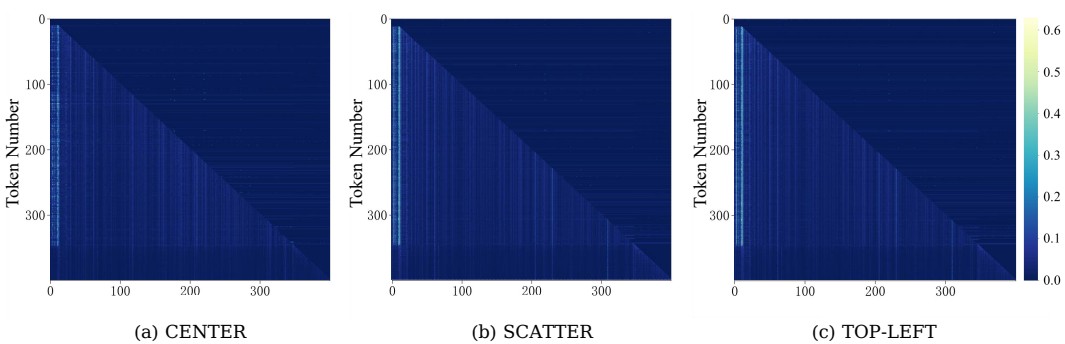

(a) CENTER          (b) SCATTER          (c) TOP-LEFT

Figure 7: **Heatmap of attention wight variation** between watermarked multimodal input sequences and original one for Qwen2.5-VL.

## A.6  Image Preprocessing for Robustness Improvement

We try to preprocess document images before entering them into VLMs to reduce the interference effect of watermarking on the model. We test a common image processing method, JPEG compression, set the compression mass to 30, and analyze its effect on watermark interference. The experimental results are shown in Figure 9.

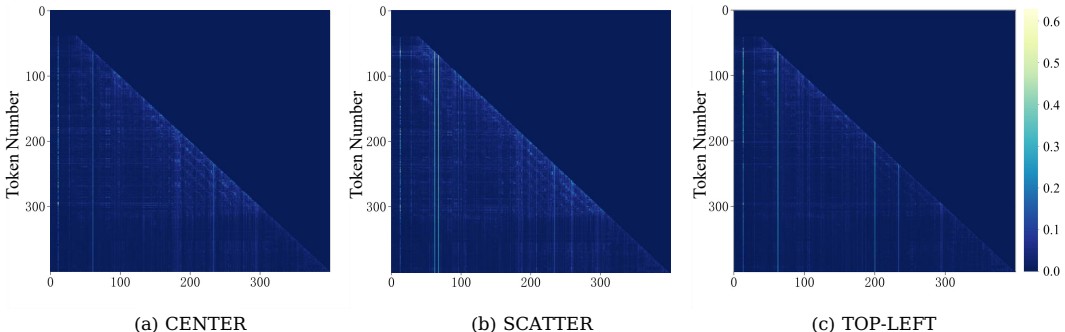

(a) CENTER          (b) SCATTER          (c) TOP-LEFT

Figure 8: **Heatmap of attention wight variation** between watermarked multimodal input sequences and original one for LLᴀVA-v1.5.

The experimental results show that the PDR value on the document data after JPEG compression is reduced compared to the unprocessed data in some cases. The main mechanism of JPEG compression is to reduce the resolution of the watermark and thus reduce its perturbation effect. However, JPEG compression is essentially a lossy compression, which not only introduces additional noise, but also reduces the overall image quality, especially affecting the sharpness of the text area. This decrease in text resolution may affect VLMs' accurate extraction and understanding of document content, resulting in an increase on PDR.

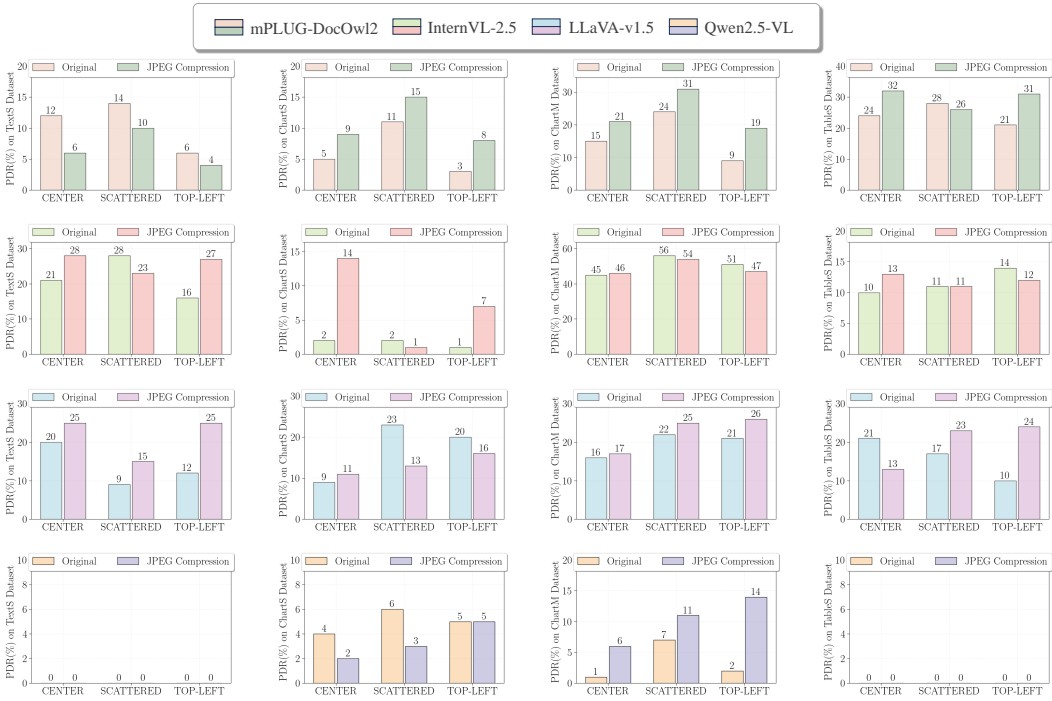

Figure 9: PDR before and after JPEG compression.

## A.7 Processing Architecture of VLMs

As show in Figure 10, before performing multimodal fusion, the model first processes the input image by dividing it into multiple smaller patches. This step allows the model to analyze local features within each patch, facilitating more effective feature extraction and representation.

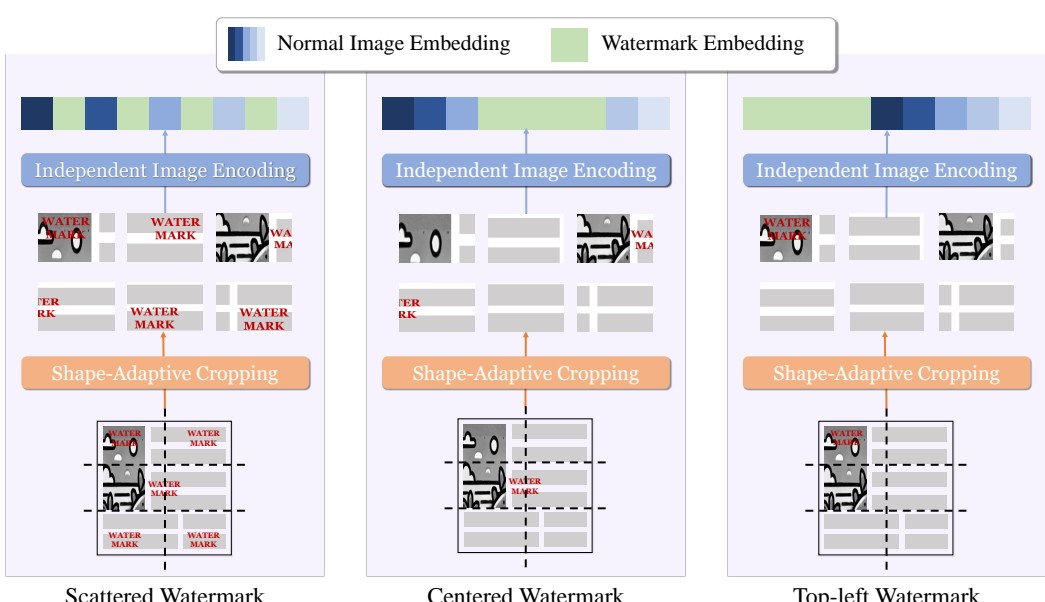

Figure 10: The structure of VLMs for processing input sequences.

