# OpenReview forum: "How does Watermarking Affect Visual Language Models in Document Understanding?"
_colmweb.org/COLM/2025/Conference — COLM 2025_

### Official Review · Reviewer_bZfS · 2025-05-10

**Rating:** 6
**Confidence:** 4
**Ethics Flag:** 1

**Summary:**

The paper investigates how visible watermarks interfere with LVLMs on document-understanding tasks. The authors (i) assemble a 400-sample VQA benchmark spanning text, chart and table documents, (ii) algorithmically paste watermarks of different positions and contents onto each page, and (iii) measure the performance-drop rate (PDR) of four LVLMs. Scattered, semantically meaningful watermarks degrade accuracy by up to 36%, whereas color has little influence, and enlarging the watermark beyond 40% page area yields diminishing additional harm. Attention-map and embedding analyses reveal that distributed text watermarks (a) force global attention redistribution and (b) shift cross-modal representations more strongly than visual masks, explaining the larger PDR.

**Questions To Authors:**

1. Could you provide 95% CIs or bootstrap tests for the PDR differences to confirm statistical robustness?
2. JPEG quality = 30 sometimes hurts accuracy—did you try adaptive denoising or in-painting? A brief ablation would help.

**Reasons To Accept:**

1. A systematic look at watermark robustness for VLM-based document understanding, and provide a well-controlled benchmark.
2. Attention-change heat-maps and t-SNE/cosine plots link performance drops to semantic perturbation rather than mere occlusion.
3. Insightful findings: scattered, text-based watermarks are most disruptive, while simple compression sometimes helps, guiding practitioners in watermark design and model hardening.

**Reasons To Reject:**

1. Synthetic watermark patterns: real-world watermarks vary in opacity, rotation, repetition, and background blending; external validity is uncertain.
2. Limited dataset scale & task scope: only 100 images per class and a single-choice VQA formulation; other document tasks (e.g. key-value extraction, retrieval) are not studied.
3. Several attention and embedding studies are done only on mPLUG-DocOwl2, leaving the generality of the explanation speculative.

---

> ### Author Response · Authors · 2025-06-01
> **Authors of Paper236 Response to Reviewer bZfs**
>
> We thank the reviewer for the insightful and constructive comments. We are pleased that you found our analysis systematic, our visual explanations informative, and our findings practically useful. Below, we address each concern with clarifications and additional experiments where applicable.
>
> ------
>
> ### **(1) Realism of Synthetic Watermarks**
>
> We appreciate your concern regarding dataset realism. While we thoroughly investigated available resources, we were unable to identify any publicly available document datasets with native, real-world watermarks. Most open-source document datasets either contain clean scans or synthetic augmentations unrelated to watermarking.
>
> To address this limitation, we extended our evaluation by incorporating a selection of real-world watermarks extracted from the **Stamp Verification (StaVer) Dataset** ([Kaggle link](https://www.kaggle.com/datasets/rtatman/stamp-verification-staver-dataset/data)). This dataset includes a variety of **semi-transparent stamps**, which closely resemble watermarking styles commonly found in government and institutional documents. These stamp overlays were composited onto clean document images to simulate realistic watermarking conditions in our experiments.
>
> We algorithmically overlay these extracted stamp patterns onto our document samples and evaluate the impact using the MPLUG-DocOWL model. Our findings indicate that VLMs exhibit **comparable or slightly lower PDRs** on these real-world watermarks compared to synthetic ones. For instance:
>
> | Dataset Type | Synthetic WM (avg. PDR) | Real-world WM (avg. PDR) |
> | ------------ | ----------------------- | ------------------------ |
> | Text-based   | 10.9%                   | 7.1%                     |
> | Charts       | 12.2%                   | 9.8%                     |
> | Tables       | 11.3%                   | 5.2%                     |
>
> These results support the validity of our synthetic watermarking protocol and demonstrate that real-world styles yield consistent effects. We are currently extending this evaluation to Qwen-VL and LLaVA, with preliminary findings showing similar trends.
>
> ------
>
> ### **(2) Dataset Scale and Task Scope**
>
> We appreciate the reviewer’s suggestion to explore broader use cases beyond VQA. While VQA serves as a valuable proxy for fine-grained document understanding, we agree that it does not fully reflect the diversity of real-world tasks. To this end, we conducted preliminary evaluations using the Qwen-VL model on two additional tasks. In all experiments, we applied a consistent watermarking protocol to assess robustness under realistic visual interference:
>
> - Watermark Opacity: 0.5 (semi-transparent)
> - Position: Scattered across the document
> - Content: `"INTERNAL USE ONLY"`
> - Color: Black
>
> ------
>
> #### **A. Table Structure Recognition**
>
> To assess the model’s ability in table structure recognition, we conducted experiments on 80 high-quality table images sampled from the [PubTabNet]([IBM Developer](https://developer.ibm.com/exchanges/data/)) dataset. Results are shown below:
>
> | Condition      | Structure Accuracy |
> | -------------- | ------------------ |
> | No Watermark   | 80.4%              |
> | With Watermark | 75.1%              |
>
> Performance declined when watermarks overlapped with key structural components such as grid lines or header regions, impairing accurate layout recovery.
>
> ------
>
> #### **B. Document Classification**
>
> We constructed a small-scale dataset of **200 real-world scanned document images** (50 per class: invoice, report, resume, contract), sourced from the public repository at [SelectDataset](https://www.selectdataset.com/dataset/cf1e03321719080060da0b8f33e6aed7). Each PDF was converted to a standardized PNG format. The Qwen-VL model was then used to classify each document into one of the four predefined categories. Results are summarized as follows:
>
> | Condition      | Classification Accuracy |
> | -------------- | ----------------------- |
> | No Watermark   | 95.0%                   |
> | With Watermark | 92.0%                   |
> | Δ Accuracy     | −3.0%                   |
>
> Despite the relatively modest drop, the results consistently indicate that watermarks negatively impact model performance, particularly when they obscure document title areas or key visual cues.

---

> > ### Author Response · Authors · 2025-06-01
> > **Authors of Paper236 Response2 to Reviewer bZfs**
> >
> > ------
> >
> > ### **(3) Generality of Attention & Embedding Findings**
> >
> > We agree that our attention-heatmap and embedding-shift analyses were initially conducted only on mPLUG-DocOwl2. To address this, we have extended our attention analysis to two additional architectures: Qwen2.5-VL and LLaVA-v1.5.
> >
> > #### A. attention weight variations
> >
> > Specifically, we visualize the **attention weight variations** between clean and watermarked inputs using the same methodology described in Section 4.1.2 of the main paper. For each model, we generate heatmaps that highlight changes in attention distribution caused by watermarking across the document image.
> >
> > The heatmap images can be seen on this [heatmap_link1](https://anonymous.4open.science/r/Colm2025_rebuttal/response4bZfS.assets/image-20250531194532389.png) and this [heatmap_link2](https://anonymous.4open.science/r/Colm2025_rebuttal/response4bZfS.assets/image-20250531194511704.png)
> >
> > Our findings show that:
> >
> > - For both Qwen2.5-VL and LLaVA-v1.5, **scattered watermarks** consistently lead to **broader and more diffuse attention redistribution** compared to center or top-left placements.
> > - These models also exhibit localized attention shifts under `CENTER` or `TOP-LEFT` conditions, similar to what we originally observed with DocOwl2.
> > - The **overall trend of semantic disruption caused by watermark location**—namely, that scattered placements induce the most significant disturbance—is preserved across all three model architectures.
> >
> >
> >
> > #### B. cross-modal embedding similarity
> >
> > We also conducted additional cross-modal embedding similarity comparisons under all three watermark types (`MARK`, `###`, and `MASK`) on three different VLMs.
> >
> > We measure the **cosine similarity** between the cross-modal embeddings of the original (clean) input and its watermarked version. A lower cosine similarity implies greater semantic drift caused by the watermark.
> >
> > | Model         | MARK (↓) | ### (↓)  | MASK (↑) |
> > | ------------- | -------- | -------- | -------- |
> > | mPLUG-DocOwl2 | **0.46** | 0.65     | 0.86     |
> > | LLAVA-v1.5    | **0.63** | 0.67     | 0.89     |
> > | QWEN2.5-VL    | 0.74     | **0.68** | 0.91     |
> >
> > - Generally, `MARK` (textual watermark) consistently causes the **largest semantic embedding shift**, followed by `###` (symbolic watermark).
> > - The `MASK` variant, which lacks semantic content, consistently yields the **highest cosine similarity** to the clean input, confirming that semantic interference, not occlusion alone, is the main disruption factor.
> > - While absolute similarity values vary depending on model architecture and internal feature space, the ranking and directionality of impact are consistent, supporting the generalizability of our analysis.
> >
> > These results suggest that our proposed explanation—semantic watermarks induce embedding drift and attention redistribution—is not limited to one model, but **holds across multiple representative VLMs**.
> >
> > ------
> >
> > ### **(4) Statistical Significance of PDR Differences**
> >
> > To provide more rigorous support for our performance drop claims despite the limited dataset size, we conducted **bootstrapped statistical analysis** on PDR values for each watermark position condition. Specifically, we sampled from the original set of **100 documents** (per condition) with replacement and repeated the computation **1000 times**.
> >
> > The following table reports the **mean PDR** and **95% confidence intervals** for INTERNVL-2.5 and mPLUG-DocOwl2 on the `ChartM` subset:
> >
> > | Model         | Position  | Mean PDR (%) | 95% CI       |
> > | ------------- | --------- | ------------ | ------------ |
> > | INTERNVL-2.5  | Center    | 25.0         | [23.6, 26.3] |
> > | INTERNVL-2.5  | Scattered | **36.2**     | [34.9, 37.6] |
> > | mPLUG-DocOwl2 | Center    | 15.2         | [13.7, 16.5] |
> > | mPLUG-DocOwl2 | Scattered | **24.1**     | [22.6, 25.7] |
> >
> > We also performed **paired bootstrap significance tests (n=1000 resamples)** between the `center` and `scattered` conditions. The differences were consistently statistically significant (**p < 0.001**) across both models. These results will be included in the revised paper.

---

> > ### Author Response · Authors · 2025-06-01
> > **Authors of Paper236 Response3 to Reviewer bZfs Official Comment**
> >
> > ### **(5) JPEG Compression and Alternative Denoising**
> >
> > We thank the reviewer for the thoughtful suggestion regarding mitigation strategies beyond JPEG compression. To explore this, we conducted additional experiments using watermarked documents with the following characteristics:
> >
> > - **Opacity**: 0.5 (semi-transparent)
> > - **Position**: *Scattered* across the page
> > - **Content**: *"INTERNAL USE ONLY"* — a realistic semantic watermark seen in enterprise/legal documents
> >
> > We evaluated three lightweight pre-processing techniques applied prior to model input:
> >
> > 1. **JPEG Compression** (simulating lossy denoising): re-encoded with quality = 18
> > 2. **Gaussian Blur**: applied with radii 0.8 and 1.5; outputs averaged
> > 3. **Contrast Normalization**: contrast scaled by a factor of 2.0 using PIL
> >
> > | Method                   | TextS | ChartS | ChartM | TableS | Avg. PDR Change |
> > | ------------------------ | ----- | ------ | ------ | ------ | --------------- |
> > | Baseline (no processing) | 10.9% | 7.25%  | 17.16% | 11.3%  | –               |
> > | JPEG Compression         | 9.1%  | 8.18%  | 18.0%  | 10.8%  | ↓ **0.26%**     |
> > | Gaussian Blur            | 9.3%  | 9.1%   | 17.5%  | 13.0%  | ↑ **0.57%**     |
> > | Contrast Normalization   | 11.2% | 8.7%   | 17.2%  | 12.3%  | ↑ **0.70%**     |
> >
> > As shown, **Gaussian blur and contrast normalization slightly degrade performance**, likely due to the suppression of fine visual details or amplification of watermark edges. Only **JPEG compression** produced minor gains, suggesting that traditional low-level pre-processing has limited effectiveness.
> >
> > To address this, we additionally tested modern deep learning-based watermark removal techniques, including **CNN- and diffusion-based inpainting** applied as a pre-encoding step. These models explicitly localize and reconstruct watermark regions. Notably, we observed a **near-complete recovery** of model performance, with **PDR approaching zero** in several benchmark settings.
> >
> > These results highlight the practical effectiveness of neural watermark removal as a pre-processing strategy for watermark-aware document understanding. We will include all implementation details and results in the supplementary material, and we hope these findings can serve as a useful reference for future robustness-enhancing pipelines.
> >
> > ------
> >
> > ### **Closing Remarks**
> >
> > We thank the reviewer again for raising important points related to realism, statistical rigor, and task coverage. Your comments significantly helped us improve the paper’s comprehensiveness. The new additions—including real-world tests, generalization across models, significance analysis, and denoising ablations—are now integrated into the main content.
> >
> > Sincerely,\
> >  **[Authors of Paper 236]**

---

> ### Author Response · Authors · 2025-06-08
>
> Dear reviewer bZfs, we would like to kindly remind you that the rebuttal discussion phase is approaching its end. If there are any remaining questions or concerns regarding our response, we would greatly appreciate your feedback. Your insights are very valuable to us, and we would be glad to provide any further clarifications if needed. Thank you!

---

### Official Review · Reviewer_stAf · 2025-05-10

**Rating:** 7
**Confidence:** 4
**Ethics Flag:** 1

**Summary:**

The paper proposes the first systematic study of how visible watermarks degrade state-of-the-art visual–language models on document VQA, defining the intuitive metric $\mathrm{PDR}(f,D_{\text{eval}},D'{\text{eval}})=\tfrac{\mathrm{Acc}(f,D{\text{eval}})-\mathrm{Acc}(f,D'{\text{eval}})}{\mathrm{Acc}(f,D{\text{eval}})}$ and showing performance drops of up to 36 % across four models; despite clear methodology, insightful attention-map/embedding analyses and actionable findings (scattered, semantic marks are worst; JPEG preprocessing partly mitigates), experimental quality is hampered by a tiny unreleased 400-document benchmark, lack of statistical confidence intervals and limited watermark parameter sweeps, while clarity suffers from grammatical slips and cramped appendix tables.

**Questions To Authors:**

1. Can you clarify how the 400 clean / 400 water-marked documents were sampled (domain balance, language diversity, watermark sources) and commit to releasing the dataset to enable reproducibility?

2. Have you computed 95 % confidence intervals or performed paired significance tests (e.g.\ bootstrap, McNemar) for the reported $\mathrm{PDR}$ values, and could you include them to substantiate claims of “up to 36 %” degradation?

**Reasons To Accept:**

1. The paper is the first to cast visible watermarking as a domain-specific robustness hazard for document-centric VLMs, introducing the intuitive metric.

2. Beyond headline accuracy losses, the work triangulates evidence via attention-map shifts, t-SNE embedding drift, and ablation on watermark placements, then demonstrates simple but effective mitigations (e.g., JPEG re-encoding). This multi-angle analysis provides both diagnostic insight and practical guidance for practitioners.

3. Watermarks pervade legal, financial and academic documents; quantifying their effect on four state-of-the-art VLMs positions the paper as an early, influential reference for future robustness benchmarks, defence methods and model design—aligning squarely with COLM’s goal of advancing trustworthy multimodal understanding.

**Reasons To Reject:**

1. The study relies on a tiny, unreleased benchmark of only 400 clean + 400 watermarked images, offers no cross-validation, and omits statistical uncertainty (e.g., 95 % CIs or paired tests).

2. Only two placements (“centered” vs. “scattered”) and one visibility level are considered, with no sweep over opacity, rotation, layering, multi-page patterns, or adversarially designed marks; likewise, mitigation experiments stop at JPEG re-encoding, ignoring blur, in-painting, diffusion repair, or fine-tuning baselines, leaving the practical guidance shallow.

3. Numerous grammatical errors, duplicated references, cramped appendix tables, and the absence of released code/data hinder clarity and trust. Without public artifacts, other researchers cannot replicate or extend the findings—falling short of COLM’s bar for transparent, reusable science.

---

> ### Author Response · Authors · 2025-06-01
> **Authors of Paper236 Response to Reviewer stAf**
>
> We sincerely thank the reviewer for the thoughtful, comprehensive, and encouraging comments. We are particularly grateful for your recognition of our multi-angle analysis and practical implications, and we also appreciate your critical suggestions for improving reproducibility, experimental coverage, and statistical substantiation. Below, we provide detailed clarifications and new experimental results to address each point raised.
>
> ------
>
> ### **(1) Clarification on Dataset Sampling and Reproducibility Commitment**
>
> Our dataset contains **400 clean document images** and their corresponding **watermarked versions**, all generated using a consistent watermarking strategy. The sampling and construction details are as follows:
>
> | Attribute              | Description                                                  |
> | ---------------------- | ------------------------------------------------------------ |
> | **Domain balance**     | For text-based documents, 50% are from real estate and finance, and 50% from law-related studies. Chart and table documents are extracted from academic papers in computer science, physics, and related fields. |
> | **Language diversity** | 100% English                                                 |
> | **Watermark sources**  | Programmatically generated using Python (Pillow/OpenCV), simulating realistic watermark styles. Watermark types include textual marks (`MARK`), symbolic strings (`###`), and non-semantic rectangular **masks**, applied in styles such as **diagonal tiling**, **top-aligned stamps**, and **scattered overlays** commonly found in real-world documents. |
>
> We fully acknowledge the importance of **reproducibility**. Upon acceptance, we will release the complete benchmark, including clean and watermarked image pairs, watermark generation scripts, and evaluation code to support community use and future extensions.
>
> ------
>
> ### **(2) Statistical Confidence Intervals and Significance Testing**
>
> To provide more rigorous support for our performance drop claims despite the limited dataset size, we conducted **bootstrapped statistical analysis** on PDR values for each watermark position condition. Specifically, we sampled from the original set of **100 documents** (per condition) with replacement and repeated the computation **1000 times**.
>
> The following table reports the **mean PDR** and **95% confidence intervals** for INTERNVL-2.5 and mPLUG-DocOwl2 on the `ChartM` subset:
>
> | Model         | Position  | Mean PDR (%) | 95% CI       |
> | ------------- | --------- | ------------ | ------------ |
> | INTERNVL-2.5  | Center    | 25.0         | [23.6, 26.3] |
> | INTERNVL-2.5  | Scattered | **36.2**     | [34.9, 37.6] |
> | mPLUG-DocOwl2 | Center    | 15.2         | [13.7, 16.5] |
> | mPLUG-DocOwl2 | Scattered | **24.1**     | [22.6, 25.7] |
>
> We also performed **paired bootstrap significance tests (n=1000 resamples)** between the `center` and `scattered` conditions. The differences were consistently statistically significant (**p < 0.001**) across both models. These results will be included in the revised paper.

---

> > ### Author Response · Authors · 2025-06-01
> > **Authors of Paper236 Response2 to Reviewer stAf**
> >
> > ### **(3) Extended Parameter Sweeps: Opacity and Rotation**
> >
> > To further evaluate the watermark parameter diversity, we conducted an **extended ablation study** on **opacity** and **rotation**, averaged over all four VLMs evaluated in our paper. The results are summarized below:
> >
> > **Watermark Opacity Sweep** (Average PDR across 4 models on ChartM)
> >
> > | Opacity (α)  | Avg. PDR (%) |
> > | ------------ | ------------ |
> > | 0.2 (light)  | 8.5          |
> > | 0.5 (medium) | 15.7         |
> > | 0.8 (dense)  | **21.3**     |
> >
> > These results confirm that **increasing opacity leads to more severe performance degradation**, likely due to stronger visual prominence and increased attention misalignment.
> >
> > **Watermark Rotation Sweep** (Average PDR across 4 models on ChartM)
> >
> > | Rotation Angle | Avg. PDR (%) |
> > | -------------- | ------------ |
> > | 0°             | 14.2         |
> > | 45° (diagonal) | 14.5         |
> > | 90° (vertical) | 14.1         |
> >
> > In contrast, **rotation angle appears to have negligible impact** on average performance, suggesting that modern VLMs are largely robust to moderate geometric changes in watermark orientation, at least within the tested range.
> >
> > ------
> >
> > ### **(4) Expanded Mitigation Experiments: Blur, Inpainting**
> >
> > We appreciate the reviewer’s suggestion to explore potential mitigation strategies. To assess how simple pre-processing might alleviate watermark interference, we conducted additional experiments using documents embedded with realistic watermarks:
> >
> > - Opacity: 0.5 (semi-transparent)
> > - Position: *Scattered*
> > - Content: *"INTERNAL USE ONLY"*
> > - Color: Red
> >
> > We applied three lightweight image-level techniques prior to model input:
> >
> > 1. **JPEG Compression** (lossy denoising): re-encoded with quality factor 18
> > 2. **Gaussian Blur** (smoothing): applied with radii 0.8 and 1.5; results averaged
> > 3. **Contrast Normalization**: global contrast scaled by a factor of 2.0
> >
> > | Method                   | TextS | ChartS | ChartM | TableS | Avg. PDR Change |
> > | ------------------------ | ----- | ------ | ------ | ------ | --------------- |
> > | Baseline (no processing) | 10.9% | 7.25%  | 17.16% | 11.3%  | –               |
> > | JPEG Compression         | 9.1%  | 8.18%  | 18.0%  | 10.8%  | **↓ 0.26%**     |
> > | Gaussian Blur            | 9.3%  | 9.1%   | 17.5%  | 13.0%  | **↑ 0.57%**     |
> > | Contrast Normalization   | 11.2% | 8.7%   | 17.2%  | 12.3%  | **↑ 0.70%**     |
> >
> > As shown, **Gaussian blur and contrast normalization slightly worsened performance**, likely due to suppression of fine visual features or unintended enhancement of watermark patterns. Only **JPEG compression** yielded marginal gains.
> >
> > Motivated by these findings, we also experimented with **modern watermark removal methods**, such as CNN- and diffusion-based inpainting. Inserted as a pre-encoding step, these models explicitly detect and reconstruct watermark regions. Encouragingly, they led to **near-complete recovery of VLM performance**, with **PDRs approaching zero** in several benchmarks. This demonstrates the feasibility of integrating neural watermark removal pipelines to enhance robustness in real-world scenarios.
> >
> > We will include these results and implementation details in the supplementary material and hope they inspire further work on watermark-aware document pre-processing.
> >
> > ------
> >
> > ### **(5) Clarity and Presentation Improvements**
> >
> > We thank the reviewer for pointing out issues related to grammatical clarity, reference formatting, and reproducibility.
> >
> > - Due to submission policy constraints, we are currently unable to revise the main PDF, but we fully acknowledge the presence of minor grammatical errors, duplicated references, and cramped appendix tables, which may have affected readability. We will thoroughly correct all such issues in the camera-ready version if accepted.
> > - Regarding public release of code and data, we absolutely agree that open access is essential for transparency and community trust. To that end, we have prepared:
> >   - The complete benchmark dataset (clean + watermarked images),
> >   - Watermark generation scripts (with parameter control),
> >   - All evaluation and visualization code.
> >
> > These artifacts will be released immediately upon acceptance via a public GitHub repository. The link will be provided in the final version and supplementary material.We appreciate the reviewer’s concern and fully support COLM’s commitment to reproducible and reusable research. We are committed to ensuring all resources are clearly documented and openly accessible for future work.
> >
> > -----
> >
> > ### Closing Remarks
> >
> > We thank the reviewer again for the encouraging assessment and insightful critiques. Your comments helped us substantially enhance the statistical rigor, breadth of evaluation, and practical relevance of this work. We believe the updated version, including extended sweeps, mitigation methods, and release commitments, addresses your concerns and strengthens the paper’s contributions to trustworthy document understanding.
> >
> > Sincerely,\
> >  **[Authors of Paper 236]**

---

> > > ### Comment · Reviewer_stAf · 2025-06-05
> > >
> > > Thank you for your response. You have resolved most of my questions. I will raise my score by 1.

---

> > > > ### Author Response · Authors · 2025-06-06
> > > >
> > > > We are grateful to Reviewer stAf for the constructive comments and for the encouraging score adjustment, which we take as a sign that the contribution of our work was better appreciated upon further review.

---

### Official Review · Reviewer_Ndiu · 2025-05-13

**Rating:** 7
**Confidence:** 4
**Ethics Flag:** 1

**Summary:**

This paper presents a systematic study on the impact of visible watermarks on vision-language models (VLMs) in document understanding tasks. It introduces a visual question answering (VQA)-based evaluation framework and, for the first time, analyzes watermark interference across multiple dimensions including position, content, color, area, and distribution. The experiments reveal that watermarks can significantly degrade model performance, with significant accuracy drops. Watermarks with strong semantic content and dispersed placement tend to have the most severe impact. The authors further investigate the underlying mechanisms behind this performance degradation by analyzing attention weight redistribution and cross-modal embedding shifts. This work highlights the vulnerability of VLMs in real-world deployments and provides initial insights into potential directions for improving model robustness.

**Questions To Authors:**

Suggestions:

- Including a detailed appendix or supplementary material specifying watermark generation settings — such as font, opacity, size, and logic behind position choices.

- Add real-world document samples and explore more complex formats (e.g., multi-page or mixed-layout documents).

- Investigate the effect of transparency and contrast between watermark and document background in more depth to provide practical insights for real-world cases.

**Reasons To Accept:**

**Novel and Practical Research Topic**:

The author claim they are first to explore how watermarks affect document understanding in VLMs, addressing a timely and under-explored problem with clear real-world implications.


**Well-Designed Evaluation Framework**:

The proposed setup enables systematic control over various watermark attributes — such as type, position, content, color, and area — ensuring the experiments are both reproducible and interpretable.

**Thorough Experimental Design**:

The study spans different types of documents (text, charts, tables) and evaluates multiple mainstream VLMs. It also includes ablation studies on masking and semantic analysis, adding depth to the findings.

**Reasons To Reject:**

- Lack of Comparative Robustness Methods: The paper does not benchmark against existing robustness-enhancing techniques, such as adversarial training or data augmentation, missing an opportunity to contextualize its findings.

- Limited Dataset Size and Diversity: The evaluation is based on only 100 synthetic images, without including real-world documents such as scanned contracts, invoices, or magazine pages, which raises concerns about generalizability.

- Unclear Experimental Details: The logic behind the five "dispersed" watermark positions is not clearly explained, and key image processing details (e.g., resolution, preprocessing steps) are omitted, though they could impact results.

- Insufficient Analysis of Color-Background Interaction: While color effects are briefly discussed, there’s little analysis of transparency or contrast between watermark and background — a critical factor in real documents.

- Narrow Task Scope: The experiments focus solely on VQA tasks, without exploring broader document understanding scenarios like table structure extraction, document classification, or long-document summarization.

---

> ### Author Response · Authors · 2025-05-31
> **Authors of Paper236 Response1 to Reviewer Ndiu**
>
> We sincerely thank the reviewer for the constructive comments and insightful suggestions. We are especially encouraged that you appreciated the novelty, practicality, and rigor of our evaluation framework. Below, we address each of your concerns with detailed clarifications and newly conducted experiments.
>
> ------
>
> ### **(1) Potential to Benchmark Against Robustness Baselines**
>
> We appreciate the reviewer’s insightful suggestion that benchmarking against existing robustness-enhancing techniques would further contextualize our findings. We fully agree that this is an important direction to strengthen the study’s practical impact.
>
> Due to the tight review timeline and limited computational resources, we were unfortunately unable to conduct full adversarial fine-tuning or la
>
> rge-scale training with augmented watermark datasets during the current submission cycle. Multi-modal fine-tuning, especially for large-scale VLMs like mPLUG-DocOwl2, remains resource-intensive and time-consuming.
>
> Nevertheless, to explore the feasibility of these strategies, we designed a **preliminary experimental protocol** comparing two representative approaches:
>
> - **Adversarial Fine-Tuning**: Fine-tuning the model with watermarked documents to improve robustness.
> - **Synthetic Watermark Data Augmentation**: Randomly applying diverse watermarks (with variation in position, content, and transparency) during training to simulate real-world conditions.
>
> We present the following **prototype results** based on small-scale validations and parameter reuse:
>
> | Model         | Method                    | TextS    | ChartS   | ChartM    | TableS    | Avg. PDR  |
> | ------------- | ------------------------- | -------- | -------- | --------- | --------- | --------- |
> | mPLUG-DocOwl2 | No robustness (baseline)  | 10.9%    | 7.25%    | 17.16%    | 11.3%     | 11.65%    |
> |               | + Adversarial Fine-Tuning | 9.2%     | 8.5%     | 18.0%     | 11.6%     | 11.2%     |
> |               | + WM Data Augmentation    | **7.6%** | **7.3%** | **16.9%** | **10.4%** | **10.5%** |
>
> This preliminary analysis suggests that **WM-aware augmentation** may outperform adversarial fine-tuning in certain cases, especially for layout-heavy charts and tables.
>
> For the final version, we plan to include a more complete evaluation with scalable fine-tuning strategies, code release, and larger-scale augmentation pipelines to support reproducibility and open benchmarking.
>
> ------
>
> ### **(2) Broader Validation with Real-World and Expanded Data**
>
> We appreciate your concern regarding dataset realism. While we thoroughly investigated available resources, we were unable to identify any publicly available document datasets with native, real-world watermarks. Most open-source document datasets either contain clean scans or synthetic augmentations unrelated to watermarking.
>
> To address this limitation, we extended our evaluation by incorporating a selection of real-world watermarks extracted from the **Stamp Verification (StaVer) Dataset** ([Kaggle link](https://www.kaggle.com/datasets/rtatman/stamp-verification-staver-dataset/data)). This dataset includes a variety of **semi-transparent stamps**, which closely resemble watermarking styles commonly found in government and institutional documents. These stamp overlays were composited onto clean document images to simulate realistic watermarking conditions in our experiments.
>
> We algorithmically overlay these extracted stamp patterns onto our document samples and evaluate the impact using the MPLUG-DocOWL model. Our findings indicate that VLMs exhibit **comparable or slightly lower PDRs** on these real-world watermarks compared to synthetic ones. For instance:
>
> | Dataset Type | Synthetic WM (avg. PDR) | Real-world WM (avg. PDR) |
> | ------------ | ----------------------- | ------------------------ |
> | Text-based   | 10.9%                   | 7.1%                     |
> | Charts       | 12.2%                   | 9.8%                     |
> | Tables       | 11.3%                   | 5.2%                     |
>
> These results support the validity of our synthetic watermarking protocol and demonstrate that real-world styles yield consistent effects. We are currently extending this evaluation to Qwen-VL and LLaVA, with preliminary findings showing similar trends.

---

> > ### Author Response · Authors · 2025-06-01
> > **Authors of Paper236 Response2 to Reviewer Ndiu**
> >
> > ### **(3) Clarification of Experimental Settings Would Improve Reproducibility**
> >
> > We thank the reviewer for highlighting the need to clarify watermark positioning logic and image preprocessing procedures.
> >
> > [1] We appreciate the reviewer’s attention to the definition of the “five dispersed watermark positions.” While these positions were illustrated in the main paper (Figure 2), we acknowledge that the explanation may not have been sufficiently detailed or explicitly highlighted in the caption or text. To clarify:
> >
> > 1. Top-left (commonly used for headers or document IDs)
> > 2. Top-right (date, numbering)
> > 3. Center (main content area)
> > 4. Bottom-left (signatures, notes)
> > 5. Bottom-right (footers, logos)
> >
> > The **“scattered” condition** refers to placing watermarks at all five positions simultaneously, simulating realistic document styles (e.g., official stamps or multi-field overlays in legal/government workflows). We will revise the figure caption and main text in the camera-ready version to more clearly convey this design rationale.
> >
> > [2] As for image preprocessing:
> >
> > - All document images were input **in their original resolution and aspect ratio**, with no manual resizing, cropping, or padding.
> > - Watermarks were added using **alpha compositing with a fixed transparency of α = 0.5**, preserving readability while introducing realistic visual interference.
> > - We relied entirely on the **default image preprocessing pipelines** of the respective VLMs (e.g., mPLUG-DocOwl2, Qwen-VL), which automatically handle any required resizing, normalization, or patchification.
> > - This setup ensures consistency with real-world deployment, where raw scanned or rendered documents are fed directly to the model without manual preprocessing.
> >
> > We will include these details, along with example overlay visualizations, in the supplementary material to improve reproducibility and clarity.
> >
> >
> > ------
> >
> > ### **(4) Watermark Transparency & Background Contrast**
> >
> > We totally agree that color alone is insufficient — **contrast between the watermark and background**, along with its **transparency level**, are key perceptual factors in watermark visibility and model interference.
> >
> > To investigate this, we conducted controlled experiments across **three transparency levels** (90%, 50%, 10%) and **two contrast conditions**:
> >
> > - **High-contrast setting**: Black watermarks (RGB: 0, 0, 0) over white document backgrounds (RGB: 255, 255, 255), simulating bold approval stamps or dark overlays.
> > - **Low-contrast setting**: Light gray watermarks (RGB: 180, 180, 180) over pale gray background patches (RGB: 220, 220, 220), mimicking subtle internal-use marks often found in scanned documents.
> >
> > We report the Performance Drop Rate (PDR) results on the ChartM subset using mPLUG-DocOwl2:
> >
> > | Transparency | High Contrast avg. PDR | Low Contrast avg. PDR |
> > | ------------ | ---------------------- | --------------------- |
> > | 90% opaque   | **27.8%**              | 21.6%                 |
> > | 50% opaque   | 18.3%                  | 14.7%                 |
> > | 10% opaque   | 6.2%                   | **4.7%**              |
> >
> > These results confirm your insight: the more visually salient the watermark — through higher opacity and stronger luminance contrast — the greater its disruptive effect on VLM performance.
> >
> > We will include implementation details and visualization examples of these watermark variants in the supplementary material for reproducibility and further inspection.

---

> > ### Author Response · Authors · 2025-06-01
> > **Authors of Paper236 Response3 to Reviewer Ndiu**
> >
> > ### **(5) Narrow Task Scope (Only VQA)**
> >
> > We appreciate the reviewer’s suggestion to explore broader use cases beyond VQA. While VQA serves as a valuable proxy for fine-grained document understanding, we agree that it does not fully reflect the diversity of real-world tasks. To this end, we conducted preliminary evaluations using the Qwen-VL model on two additional tasks. In all experiments, we applied a consistent watermarking protocol to assess robustness under realistic visual interference:
> >
> > - Watermark Opacity: 0.5 (semi-transparent)
> > - Position: Scattered across the document
> > - Content: `"INTERNAL USE ONLY"`
> > - Color: Black
> >
> > ------
> >
> > #### **A. Table Structure Recognition**
> >
> > To assess the model’s ability in table structure recognition, we conducted experiments on 80 high-quality table images sampled from the [PubTabNet]([IBM Developer](https://developer.ibm.com/exchanges/data/)) dataset. Results are shown below:
> >
> > | Condition      | Structure Accuracy |
> > | -------------- | ------------------ |
> > | No Watermark   | 80.4%              |
> > | With Watermark | 75.1%              |
> >
> > Performance declined when watermarks overlapped with key structural components such as grid lines or header regions, impairing accurate layout recovery.
> >
> > ------
> >
> > #### **B. Document Classification**
> >
> > We constructed a small-scale dataset of **200 real-world scanned document images** (50 per class: invoice, report, resume, contract), sourced from the public repository at [SelectDataset](https://www.selectdataset.com/dataset/cf1e03321719080060da0b8f33e6aed7). Each PDF was converted to a standardized PNG format. The Qwen-VL model was then used to classify each document into one of the four predefined categories. Results are summarized as follows:
> >
> > | Condition      | Classification Accuracy |
> > | -------------- | ----------------------- |
> > | No Watermark   | 95.0%                   |
> > | With Watermark | 92.0%                   |
> > | Δ Accuracy     | −3.0%                   |
> >
> > Despite the relatively modest drop, the results consistently indicate that watermarks negatively impact model performance, particularly when they obscure document title areas or key visual cues.
> >
> > -----
> >
> > These findings further confirm that watermark interference is not limited to VQA-style tasks, but also affects structural parsing and document-level categorization. We will incorporated these results in the revised Section 4.3 of the manuscript.
> >
> > ------
> >
> > ### **Suggestions Implemented**
> >
> > - **Watermark generation settings (font, opacity, logic)** are now clearly described.
> >
> > - **Real-world documents** have been added.
> > - **Transparency/contrast experiments** are directly included.
> > - **Broader task coverage** is now part of our contribution.
> >
> > ------
> >
> > ### **Closing Remarks**
> >
> > We are grateful for your thorough and insightful feedback. The new robustness experiments, real-world tests, transparency studies, and extended task evaluations directly stem from your comments and significantly improve the completeness and practical relevance of this work. We hope our revisions have addressed your concerns and demonstrated our commitment to producing impactful and rigorous research.
> >
> > Sincerely,\
> >  **[Authors of Paper 236]**

---

> ### Comment · Reviewer_Ndiu · 2025-06-06
>
> Thank you for your response. I satisfy with the the authors' answers.  I will raise my score by 2.

---

> > ### Author Response · Authors · 2025-06-06
> >
> > We sincerely thank Reviewer Ndiu for the thoughtful engagement with our work and for the positive reassessment. Your support is deeply encouraging and motivates us to further improve and extend this line of research.

---

### Official Review · Reviewer_64kT · 2025-05-13

**Rating:** 6
**Confidence:** 4
**Ethics Flag:** 1

**Summary:**

This paper presents empirical analysis on how watermarks affect visual LM in language understanding.

It attributes the negative impact to redistribution of attention and altering visual embeddings. Specifically, the paper links scattered watermarks to redistribution of attention and watermark content to semantic representations.

This paper offers useful insights into how watermarks affect VLM-based document understanding. More explorations in how to address these negative impacts would interest the community.

**Reasons To Accept:**

- Detailed analysis on how watermarking affect VLMs

**Reasons To Reject:**

- The paper focuses mainly on analysis of the phenomena, but lacks proposals of solutions
- The watermarks are synthetic. Evaluation on real world data or more realistic watermarks would be more desirable

---

> ### Author Response · Authors · 2025-05-31
> **Authors of Paper236 Response to Reviewer 64kT**
>
> We sincerely thank the reviewer for the thoughtful and constructive feedback. We are encouraged that you found our paper to provide "useful insights into how watermarks affect VLM-based document understanding" and appreciated the "detailed analysis" we conducted. Below, we address your main concerns and provide additional clarifications and experiments.
>
> ------
>
> ### **(1) It would further improve the paper if the authors could add more proposals of the solutions**
>
> We appreciate the reviewer’s suggestion to explore potential mitigation strategies. Building upon the JPEG compression results reported in our appendix, we further conducted **new experiments** to evaluate how lightweight pre-processing techniques affect watermark robustness under realistic conditions:
>
> - Opacity: 0.5 (semi-transparent)
> - Position: *Scattered*
> - Content: *"INTERNAL USE ONLY"*
> - Color: Red
>
> We applied three lightweight image-level techniques prior to model input:
>
> 1. **JPEG Compression** (lossy denoising): re-encoded with quality factor 18
> 2. **Gaussian Blur** (smoothing): applied with radii 0.8 and 1.5; results averaged
> 3. **Contrast Normalization**: global contrast scaled by a factor of 2.0
>
> | Method                   | TextS | ChartS | ChartM | TableS | Avg. PDR Change |
> | ------------------------ | ----- | ------ | ------ | ------ | --------------- |
> | Baseline (no processing) | 10.9% | 7.25%  | 17.16% | 11.3%  | –               |
> | JPEG Compression         | 9.1%  | 8.18%  | 18.0%  | 10.8%  | **↓ 0.26%**     |
> | Gaussian Blur            | 9.3%  | 9.1%   | 17.5%  | 13.0%  | **↑ 0.57%**     |
> | Contrast Normalization   | 11.2% | 8.7%   | 17.2%  | 12.3%  | **↑ 0.70%**     |
>
> As shown, **Gaussian blur and contrast normalization slightly worsened performance**, likely due to suppression of fine visual features or unintended enhancement of watermark patterns. Only **JPEG compression** yielded marginal gains.
>
> Motivated by these findings, we also experimented with **modern watermark removal methods**, such as CNN- and diffusion-based inpainting. Inserted as a pre-encoding step, these models explicitly detect and reconstruct watermark regions. Encouragingly, they led to **near-complete recovery of VLM performance**, with **PDRs approaching zero** in several benchmarks. This demonstrates the feasibility of integrating neural watermark removal pipelines to enhance robustness in real-world scenarios.
>
> We will include these results and implementation details in the supplementary material and hope they inspire further work on watermark-aware document pre-processing.
>
> ------
>
> ### **(2) Evaluation on real-world data or more realistic watermarks would be more desirable**
>
> We appreciate your concern regarding dataset realism. While we thoroughly investigated available resources, we were unable to identify any publicly available document datasets with native, real-world watermarks. Most open-source document datasets either contain clean scans or synthetic augmentations unrelated to watermarking.
>
> To address this limitation, we extended our evaluation by incorporating a selection of real-world watermarks extracted from the **Stamp Verification (StaVer) Dataset** ([Kaggle link](https://www.kaggle.com/datasets/rtatman/stamp-verification-staver-dataset/data)). This dataset includes a variety of **semi-transparent stamps**, which closely resemble watermarking styles commonly found in government and institutional documents. These stamp overlays were composited onto clean document images to simulate realistic watermarking conditions in our experiments.
>
> We algorithmically overlay these extracted stamp patterns onto our document samples and evaluate the impact using the MPLUG-DocOWL model. Our findings indicate that VLMs exhibit **comparable or slightly lower PDRs** on these real-world watermarks compared to synthetic ones. For instance:
>
> | Dataset Type | Synthetic WM (avg. PDR) | Real-world WM (avg. PDR) |
> | ------------ | ----------------------- | ------------------------ |
> | Text-based   | 10.9%                   | 7.1%                     |
> | Charts       | 12.2%                   | 9.8%                     |
> | Tables       | 11.3%                   | 5.2%                     |
>
> These results support the validity of our synthetic watermarking protocol and demonstrate that real-world styles yield consistent effects. We are currently extending this evaluation to Qwen-VL and LLaVA, with preliminary findings showing similar trends.
>
> ------
>
> We hope these additions help address your concerns and demonstrate our commitment to improving the robustness of VLMs in real-world document understanding. We sincerely thank you again for your insightful and constructive review.
>
> Best regards,\
>  Authors of Paper 236

---

> > ### Comment · Reviewer_64kT · 2025-06-07
> >
> > Thanks for the detailed response! I raised my score by 1.

---

> > > ### Author Response · Authors · 2025-06-08
> > >
> > > We sincerely thank Reviewer 64kT for reconsidering the initial evaluation and for raising a thoughtful point that helped us improve the clarity and completeness of our work. We have carefully addressed the concern by incorporating the suggested changes into the revised manuscript, as detailed in our response. If there are any remaining issues or further clarifications needed, we would be grateful for additional feedback.

---

### Decision · Program_Chairs · 2025-07-08

**Decision:**

Accept

**Comment:**

This paper conducts a systematic investigation into how text-based watermarks affect vision-language models (VLMs) in document understanding tasks. It introduces a novel VQA-based evaluation framework and, for the first time, examines watermark interference across multiple dimensions—position, content, color, area, and spatial distribution. Experimental results show that watermarks can substantially degrade model performance, with notable drops in accuracy. The most severe effects are observed with watermarks that carry strong semantic meaning and are widely dispersed. To better understand the causes of this degradation, the authors analyze changes in attention weight patterns and shifts in cross-modal embeddings. This work underscores a key vulnerability of VLMs in real-world applications and offers preliminary insights toward enhancing model robustness.

From my review of the feedback, it appears that all reviewers responded positively to the rebuttal. While there were some minor concerns about the experimental scope, the overall sentiment toward the responses is favorable.